# Solid-State Fermentation with White Rot Fungi (*Pleurotus* Species) Improves the Chemical Composition of Highland Barley Straw as a Ruminant Feed and Enhances In Vitro Rumen Digestibility

**DOI:** 10.3390/jof9121156

**Published:** 2023-11-30

**Authors:** Yuqiong Wang, Changlong Gou, Liming Chen, Yangci Liao, Hang Zhang, Lilong Luo, Jiahang Ji, Yu Qi

**Affiliations:** 1College of Animal Science and Technology, Inner Mongolia Minzu University, Tongliao 028000, China; wangyuq2015@163.com (Y.W.); chenliming1215@163.com (L.C.); hang0523@163.com (H.Z.); luolilong0521@163.com (L.L.); jjh001103@163.com (J.J.); wsqiyu@163.com (Y.Q.); 2Institute of Pratacultural, Tibet Academy of Agricultural and Animal Husbandry Sciences, Lhasa 850000, China; lyangci@taaas.org

**Keywords:** highland barley straw, *Pleurotus* spp., white rot fungi, lignin, in vitro digestibility

## Abstract

Lignin degradation is important for enhancing the digestibility and improving the nutritive quality of ruminant feeds. White rot fungi are well known for their bioconversion of lignocellulosic biomass. The objective of this paper was to evaluate whether *Lentinus sajor-caju*, *Pleurotus ostreatus*, *Phyllotopsis rhodophylla*, *Pleurotus djamor*, *Pleurotus eryngii*, and *Pleurotus citrinopileatus* treatments altered the chemical compositions of highland barley straw constituents and enhanced their nutritional value as a ruminant feed. All white rot fungi significantly increased the relative crude protein (CP), ethyl ether extract (EE), starch, soluble protein (SP), and non-protein nitrogen (NPN) contents but decreased the ash, neutral detergent fiber (NDF), acid detergent fiber (ADF), acid detergent lignin (ADL), and acid detergent insoluble protein (ADFIP) contents. In addition, *L. sajor-caju* treatment increased (*p* < 0.001) the levels of PA, PB_2_, PB_3_, CA, CB_1_, CB_2_, and CNSC, but reduced (*p* < 0.001) the PC and CC in the solid-state fermentation of highland barley straw. Maximum ligninlysis (50.19%) was optimally produced in the presence of 1.53% glucose and 2.29% urea at 22.72 ℃. The in vitro dry matter digestibility and total volatile fatty acid concentrations of fermented highland barley straw, as well as the fermentability, were optimized and improved with *L. sajor-caju*, which degraded the lignocellulose and improved the nutritional value of highland barley straw as a ruminant feed.

## 1. Introduction

As an agriculture-based society, China produces a wide variety of crops in substantial amounts, generating enormous agricultural residues. Highland barley, one of the most widely cultivated grain crops in the Qinghai-Tibet region of China, accounts for 43% of the total grain production [1]. However, most highland barley straw is not effectively used and is discarded or burned, wasting resources and causing environmental pollution. Highland barley straw has high lignocellulose content and low protein content; therefore, it is a forage feed with limited nutritional value for ruminants. To improve straw quality, there are three main methods: physical, chemical, and biological treatments [2], with the last regarded as economical, harmless, and environmentally friendly compared to the other two [2,3].

Lignin degradation is important for enhancing the digestibility and improving the nutritive quality of ruminant feeds [4]. White rot fungi are well known for their bioconversion of lignocellulosic biomass, and they may selectively degrade lignin and release cellulose and hemicellulose under solid-state fermentation [5]. Selective white rot fungi are effective at degrading lignin [6] and improving the digestibility of ruminant feeds [7]. In addition, the growth of fungal mycelium in solid-state fermentation increases the total protein content of fermentation substrates [8,9], creating the potential to produce a cost-effective protein source. However, the excessive consumption of fermentable (not lignin) carbohydrates or slow fermentation by white rot fungi are major limitations. Thus, it is necessary to screen high-potency white rot fungal species to optimize utilization of highland barley straw as a ruminant feed.

Based on the limited nutritional value of highland barley straw, these experiments were designed to screen high-potency white rot fungal strains for degrading lignin in highland barley straw, characterize the biochemical changes in highland barley straw constituents, and enhance in vitro digestibility. 

## 2. Materials and Methods

### 2.1. Fungal Strains and Spawn Preparation

Six white rot fungi, *Lentinus sajor-caju* (Fr.) Fr. (strain CGMCC 5.592), *Pleurotus ostreatus* (Jacq.) P. Kumm. (strain CGMCC 5.729), *Phyllotopsis rhodophylla* (Bres.) Singer (strain CGMCC 5.363), *Pleurotus djamor* (Rumph. ex. Fr.) Boedijn (strain GMCC 5.600), *Pleurotus eryngii* (DC) Quél. (strain CGMCC 5.732), and *Pleurotus citrinopileatus* Singer (strain CGMCC 5.244) were obtained from the China General Microbiological Culture Collection Center (CGMCC) in Beijing, China. They were individually grown in potato dextrose agar medium (PDA) (potato 200 g, peptone 10 g, glucose 20 g, and agar 18 g per L) and stored at 4 °C for later use. Agar plates were prepared using PDA, inoculated with a 0.5 cm^2^ piece of fungus, and incubated at 25 °C for 7 days. Four agar plugs (with diameters of 8 mm) of active mycelium from a PDA plate were transferred aseptically into 250 mL Erlenmeyer flasks containing 80 mL of autoclaved potato extract dextrose broth medium (PDB) (potato 200 g, peptone 10 g, and glucose 20 g per L). The cultures were incubated at 25 °C on rotary shakers (150 rpm). 

### 2.2. Experimental Set-Up

Highland barley straw was collected from Mozhu Gongka County, Tibet, China (29°50′ N, 91°45′ E) and chopped into lengths of 1–2 cm, and distilled water was added to reach a moisture level of 65%. The substrates also contained 1% corn meal, 1%, urea, 0.5% gypsum powder, and 1% lime powder. A cultivation substrate was put into an autoclavable plastic bag, sterilized at 121 °C for 1 h, and then cooled at room temperature. The samples (50 g dry weight of highland barley straw) were inoculated with 10% spawn and incubated for 21 days in a climatic chamber at 25 ± 0.5 °C with 70–80% relative humidity. The containers with inoculated highland barley straw had six duplicate samples. The controls were kept under the same experimental conditions, but they were uninoculated. 

### 2.3. Chemical Analyses

After 21 days of incubation, the substrate samples were dried at 60 °C until they reached a constant weight. The crude protein (CP), insoluble neutral detergent protein (NDIP), and acid detergent insoluble protein (ADIP) contents were determined by measuring the nitrogen contents (AOAC, 2000; method 990.06) [10] and using a conversion factor of 6.25. The ethyl ether extract (EE) was determined (AOAC, 2000; method 2003.05). The ash contents were determined by ashing at 550 °C in a muffle furnace for 3 h. The neutral detergent fiber (NDF), acid detergent fiber (ADF), and acid detergent lignin (ADL) were assessed by slightly modifying the method used by Goering and Vansoest [11] and Van Soest [12]. The samples (0.5–1 g) were placed into polyester mesh bags (Ankom F57) and sealed; these bags and 2000 mL of neutral detergent were put into an ANKOM 200i semi-automatic fiber analyzer (ANKOM Technology Corp, Fairport, NY, USA) at 100 °C for 60 min. Then, the bags were washed to neutral with distilled water, dried, and weighed. The dried residue represents NDF. The remaining samples and 2000 mL of acid detergent were put into the semi-automatic fiber analyzer at 100 °C for 60 min, then the bags were washed to neutral with distilled water, dried, and weighed. The dried residue represents ADF, whereas the loss represents hemicellulose. The dried residue was soaked in 72% (*v*/*v*) H_2_SO_4_ and kept at 25 °C for 2 h. Thereafter, the bags were washed to neutral with distilled water, dried, and weighed and the loss represents cellulose. The remaining samples were kept at 550 °C for 3 h in a tared crucible and reweighed to calculate the loss as ADL. The soluble protein (SP) content was determined by the method used by Licitra et al. [13]. The starch content and non-protein nitrogen (NPN) were determined by AACC methods [14].

### 2.4. Protein and Carbohydrate Fractionation

The Cornell Net Carbohydrate and Protein System (CNCPS) was used to fractionate and characterize proteins and carbohydrates. The PA (rapidly degraded protein fraction), PB_1_ (true protein fast-degradable fraction), PB_2_ (medium-degradable protein fraction), PB_3_ (low-degradable protein fraction), PC (undegradable protein fraction), CA (sugars), CB_1_ (starch), CB_2_ (carbohydrates and pectin), and CC (cell walls and lignin) of the rations were analyzed and calculated according to Sniffen et al. [15].

### 2.5. Electron Microscopy for Structural Characterization of Fungal Fermented Highland Barley Straw

Scanning electron microscopy (SEM) examinations of fermented highland barley straw samples were performed at the Electron Microscope Facility. The small pieces (2 × 2 mm) of fungal-fermented highland barley straw were cut, fixed in 2.5% glutaraldehyde, dehydrated in an ascending gradient of acetone, and critical-point dried. All samples were imaged with a scanning electron microscope (MIRA3, TESCAN, Brno, Czech Republic).

### 2.6. Response Surface Methodology (RSM)

The above-described experiments identified a superior species of white rot fungus. Three variables (temperature, glucose, and urea) were varied to enhance ligninolysis. The optimization of selected variables was performed with response methodology (RSM) using a Box–Behnken design (Box and Behnken, 1960) [16]. Each variable was studied at three levels (−1, 0, and 1). The experimental design included a 17-fermentation package with 5 central points. Each fermentation package contained 50 g of highland barley straw, 1–2% glucose, and/or 1–3% urea in a climatic chamber at 15–29 °C (Table 1).
(1)$$G=\beta_{0}\;+\;\beta_{1}X_{1}\;+\;\beta_{2}X_{2}\;+\;\beta_{3}X_{3}\;+\;\beta_{11}{X^{2}}_{1\;}+\;\beta_{22}{X^{2}}_{2\;}+\;\beta_{33}{X^{2}}_{3\;}+\;\beta_{12}X_{1}X_{2\;}+\;\beta_{13}X_{1}X_{3\;}+\;\beta_{23}X_{2}X_{3\;}$$
where G is the predicted response; β_0_, intercept; β_1_, β_2_, and β_3_, linear coefficients; β_11_, β_22_, and β_33,_ squared coefficients and β_12_, β_13_ and β_23_, interaction coefficients; and the independent variables X_1_, X_2_, and X_3_ were calculated via the quadratic model (Equation (1)). MINITAB and the statistical software package Design Expert^®^ Version 8.0 (State-Ease, Inc., Minneapolis, MN, USA) were used to obtain optimal conditions and generate response surface graphs.

### 2.7. In Vitro Fermentation

The in vitro dry matter digestibility (IVDMD) and rumen fermentation parameters of samples were estimated according to Wang et al. [17]. Fresh rumen fluid was collected in the morning from three yaks fed highland barley straw and concentrate (ratio of 60:40 [dry matter]). The samples (0.6 g) were placed into a 120 mL fermentation bottle under continuous flushing with CO_2_. Every fermentation bottle received 50 mL of buffered rumen fluid and was incubated for 72 h at 39 °C. After 48 h of incubation at 39 °C, the samples were analyzed for IVDMD, and the rumen fluid was analyzed for pH, volatile fatty acids (VFA), and NH_3_-N.

The rumen fluid was filtered through six layers of cheesecloth and the pH of the filtered fluid was measured immediately with a PHS-3C pH meter (Olabo, Jinan, Shandong, China). Culture fluid from each sample (~10 mL) was centrifuged at 10,000× *g* for 15 min at 4 °C. The supernatants were acidified with 25% (*w*/*v*) metaphosphoric acid in a 1:9 acid:rumen fluid ratio, and concentrations of VFAs were determined via gas chromatography (Agilent 7890A, Agilent Inc., Palo Alto, CA, USA) as described by Wang et al. [18]. Briefly, the VFA were detected using an R flame ionization detector (FID) after acid separation, with nitrogen gas at a flow rate of 0.8 mL min^−1^. After being isothermal for 2 min at 60 °C, the temperature was increased to 220 °C at a rate of 20 °C min^−1^, with a detector temperature of 280 °C. Thereafter, the VFA were identified and quantified from the chromatograph peak areas using calibration with external standards. The ruminal NH_3_-N concentrations were measured as described by Weatherburn [19]. The microbial protein nitrogen was measured via Kjedahl distillation after differential centrifugation (AOAC, 2000; method 990.06) [10]. All analyses were repeated in three replicates.

### 2.8. Statistical Analyses

All data were analyzed using the general linear model procedure (GLM), followed by Duncan’s multiple range tests using Statistical Analysis System (SAS Version 9.0; SAS Institute Inc., Cary, NC, USA). The means were separated using least square means and presented with standard errors of the mean (SEM). All data were analyzed using the following statistical model: Yij =μ+αi+εij
where *Yij* = the response variable, *μ* = the general mean, *αi* = the effect of white rot fungi, and *εij* = the random error. The results were considered different when *p* ≤ 0.05.

## 3. Results and Discussion

### 3.1. Chemical Composition

The chemical compositions (dry matter (DM) basis) of highland barley straw cultured with white rot fungi are shown in Table 2. The CP, EE, starch, SP, and NPN contents following white rot fungi treatment were higher than in controls (*p* < 0.001), whereas the ash, NDF, ADF, ADL, and ADFIP contents following white rot fungi treatment were decreased compared to controls (*p* < 0.001). The NDFIP contents of most fungi treatments (except *P. citrinopileatus*) were decreased (*p* < 0.001) compared to controls. The treatment of highland barley straw with *L. sajor-caju* maximized CP, EE, starch, SP, and NPN (44.57, 40.09, 1.26, 87.90, and 120.9% DM), with maximally decreased NDF, ADF, ADL, NDFIP, and ADFIP (26.21, 46.29, 20.79, 35.86, and 62.90%) compared to controls. Of all white rot fungi, *L. sajor-caju* treatment had the best results for relatively improving the chemical composition.

White rot fungi have important roles in the biodegradation of lignin, making them suitable for the delignification of roughage. Fungal treatment has improved the nutritional value of various crop stalks, including the following kinds of straw: rice, corn, wheat, paddy, and rape [19]. Therefore, white rot fungi are an efficient treatment for a wide range of agricultural byproducts. In the present study, the selection of highland barley straw fermentation was verified. The CP contents increased with white rot fungi treated for 21 days, which is consistent with studies of Nie et al. [20]. White rot fungi break down organic material to obtain carbon and nitrogen to support fungal growth and development [21]. Some organic material is converted to CO_2_ and H_2_O, with an increase in CP contents and reduction in lignocellulose, which is consistent with a previous report [22]. In another study, six strains of white rot fungi were evaluated on corn stover and *L. sajor-caju* had the highest lignin reduction (degraded 38.29%) after 30 d of fungal pretreatment [23]. In addition, the best fungi for selectively removing lignin from stored beech logs were *L. sajor-caju* and *P. ostreatus* (losses of 29.92 and 27.42%, respectively) [24]. Therefore, *P. ostreatus* was inferior to *L. sajor-caju* for degrading the lignin. This is partially consistent with our findings. 

### 3.2. Protein and Carbohydrate Fractionation

The CNCPS protein fractions of highland barley straw with white rot fungi are shown in Table 3. The PA fractions of highland barley straw were all increased (*p* < 0.001) after incubation with fungi, with the highest levels in *L. sajor-caju* treatment (22.65% of DM). The PB_1_ fraction of *P. djamor* treatment was decreased (*p* < 0.001), but with the exception of *P. eryngii* treatment, the other four fungi caused increases (*p* < 0.001). The PB_2_ and PB_3_ fractions of *L. sajor-caju* and *P. ostreatus* treatments were increased (*p* < 0.001) compared to all other treatments. The PC fractions of highland barley straw were all decreased (*p* < 0.001) after incubation with fungi, but the lowest levels followed *L. sajor-caju* treatment (7.29% of DM).

The CNCPS carbohydrate fractions of highland barley straw with white rot fungi are shown in Table 4. The levels of CHO fractions of *P. rhodophylla* and *P. djamor* treatments were significantly increased, but *L. sajor-caju* treatment had the opposite result (*p* < 0.001). The levels of CA, CB_1_, CB_2_, and CNSC fractions after *L. sajor-caju* treatment were higher than for other treatments and the controls (*p* < 0.001), but the levels of CC fraction were significantly lower. With the exception of *L. sajor-caju* treatment, the levels of CB_2_ fractions were not affected compared to the controls (*p* > 0.05).

The Cornell Net Carbohydrate and Protein System is widely applied and internationally recognized as a dynamic model for feed evaluation [13]. The application of CNCPS reflects not only the carbohydrate compositions but also potential rumen fermentative activity [19]. The PB_2_ fraction is the largest protein pool and includes two-thirds of NPN [25,26]; similarly, in our data, the amount of PB_2_ in highland barley straw treated with each of the six fungi was higher than those of other CNCPS protein fractions. The levels of PB_2_ and PB_3_ fractions are closely related to the amount of rumen protein. In the present study, the levels of PA, PB_1_, PB_2_, and PB_3_ fractions were all increased by the used white rot fungi. Furthermore, increases in rumen protein contents and the efficiency of protein digestion and utilization of highland barley straw were optimized by *L. sajor-caju* treatment. Treatment of highland barley straw with white rot fungi reduced the PC contents, with a maximal decrease induced by *L. sajor-caju* (loss of 73.91%), reflecting protein bioavailability. In contrast, Nie [19] reported that the lignin matrix was not as well decomposed as the PC.

### 3.3. Electron Microscopy for Structural Characterization of Fungally Fermented Highland Barley Straw

The SEM morphology of highland barley straw before and after pretreatments is shown (Figure 1). Whereas unfermented highland barley straw had structures that were intact and clear, fermented highland barley straw had a rough texture and surface pores of variable diameter. Furthermore, numerous white rot fungi mycelia covered the surface of highland barley straw after 21 days of fermentation. The loosening of morphological structure stem barks fermented with *L. sajor-caju* was more obvious than for the unfermented samples, with almost the entire surface fractured. These results further confirm the deep fungal penetration and progressive degradation of the highland barley straw. The straw microstructure was clear, similar to reports during biological upgrading under solid state fermentation conditions by white rot fungi [27,28], and serves as morphological evidence that white rot fungi relatively enhances the nutritional value of ruminant feed.

### 3.4. Response Surface Methodology (RSM)

Changes in cultivation factors (glucose and urea concentrations as well as temperature) changed fiber degradation (Table 5, Figure 2, Figure 3 and Figure 4).

These data were analyzed by applying a multiple regression analysis method based on Equation (2). The predicted responses for lignin degradation (G) were as follows:(2)$$G=49.92+1.852X_{1}+0.175X_{2}+1.17X_{3}-0.985{X^{2}}_{1}-0.43{X^{2}}_{2}+1.08{X^{2}}_{3}-8.155X_{1}X_{2}-3.655X_{1}X_{3}-2.02X_{2}X_{3}$$

The interactive effects among glucose, urea, temperature (*X*_1_*X*_2_, *X*_1_*X*_3_, *X*_2_*X*_3_), linear (*X*_2_), squared (*X*^2^_2_, *X*^2^_3_) and interactive (*X*_2_*X*_3_) were all very significant for lignin degradation (*p* < 0.001). The maximum ligninlysis (50.19%) of highland barley straw after 21 days was caused by *L. sajor-caju* and 1.52% glucose and 2.29% urea at 22.72 °C.

The lignin decomposition in white rot fungi is highly regulated by nutrients and conditions (nutrient contents and temperature) [29]. Nitrogen and carbon are both important for fungi growth and stimulate ligninolytic capacity [30]. Appropriate nitrogen supplementation can enhance lignin degradation, whereas a high concentration probably decreases lignin degradation [31], which is consistent with the present study. As a supplemental nitrogen source to promote lignin degradation, 2% urea was superior to 3%. Concentrations of carbon sources in the fungal medium can have either positive or negative effects on the fungus. During *L. sajor-caju* fermentation, the supplementation of glucose increases lignin degradation, which is consistent with the findings of Sarria-Alfonso et al. [32]. However, the effects of carbon source on lignocellulosic substrates depend not only on concentrations but also fungal species. Fortina et al. [33] found that the supplementation of 2% glucose in *Botrytis cinerea* media effectively stimulated lignin degradation. Similarly, the other two factors also significantly affected mycelium production in the present study, emphasizing the need to the optimize fungal fermentation conditions.

### 3.5. In Vitro Rumen Fermentation

The in vitro fermentation products of control intact highland barley straw and highland barley straw pretreated with *L. sajor-caju* are shown in Table 6. Except for pH, the ttreatment of highland barley straw with *L. sajor-caju* affected (*p* < 0.001) all other end points compared to the controls. Furthermore, for this fungus, propionic acid was lower than in both the controls (*p* < 0.001). However, all other fermentation products were the opposite.

The IVDMD is directly associated with degree of lignification or delignification of roughage and is an assessment of nutritional quality [34]. Treatment with *L. sajor-caju* increased IVDMD and therefore substantially reduced lignin compared to controls. White rot fungi had similar effects on rice straw, as reported by Zheng et al. [35]. The pH is an important indicator of rumen environment and should be >6.3 for optimal rumen metabolism. The rumen pH was not affected by *L. sajor-caju* treatment of highland barley straw in the present study, which is consistent with the findings of Datsomor et al. [22]. The production of VFA by carbohydrate fermentation constitutes 70–80% of ruminant energy reserves and reflects feed digestibility. In the present study, *L. sajor-caju* treatment of highland barley straw increased total VFA compared to the controls. Perhaps this was due to total VFA concentrations being positively correlated with IVDMD; with higher digestibility, more carbohydrates are available for rumen microbes. Similarly, corn straw treatment with *P. ostreatus* yielded greater total VFA concentrations compared to the controls [36].

Acetate is the primary VFA produced by microbes and is an important energy source for ruminants. Ruminal acetic acid concentrations were positively correlated with the degree of fermentation of cellulose and hemicellulose [37]. In the present study, acetic acid concentrations relatively increased with *L. sajor-caju* treatment of highland barley straw compared to the controls, attributed to *L. sajor-caju* promoting lignin degradation and increasing the utilization of cellulose and hemicellulose. The production of ammonia-N during fermentation is closely related to the degradation of dietary nitrogen and absorption of nitrogen by rumen bacteria [38]. The high concentrations of ammonia-N in the *L. sajor-caju* treatment of highland barley straw were attributed to the formation of mycelial protein, which increased the digestion and utilization of protein compared to the controls. Microbial protein synthesis depends largely on fermentable and available DM and nitrogen in the rumen. Increased microbial protein nitrogen synthesis may have been due to increased fiber digestion by *L. sajor-caju*, which stimulated ammonia-N utilization.

## 4. Conclusions

In conclusion, the investigated white rot fungi significantly degraded lignocellulose and increased the CP content of fermented highland barley straw. Based on the nutritional value of fermented highland barley straw, *L. sajor-caju* had the greatest ability to degrade lignin and improve the nutritional quality. The optimal concentrations of carbon and nitrogen and temperature for solid state fermentation increase the lignin degradation rate of in *L. sajor-caju* by 3.9%. Finally, fermenting highland barley straw with *L. sajor-caju* increased in vitro digestibility and altered VFAs.

## Figures and Tables

**Figure 1 jof-09-01156-f001:**
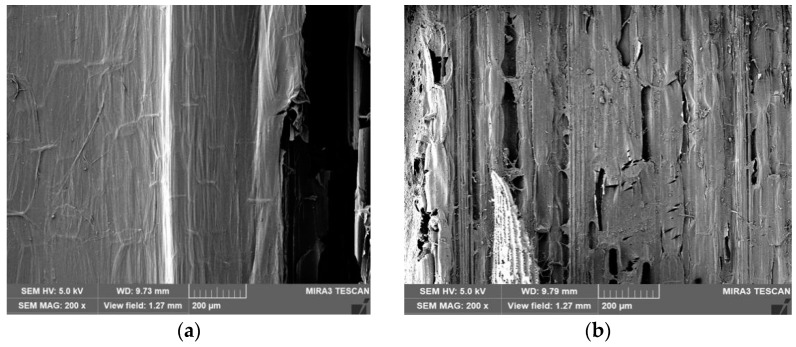
Scanning electron micrograph of unfermented ((**a**) 200 μm) and *L. sajor-caju* fermented highland barley straw ((**b**) 200 μm).

**Figure 2 jof-09-01156-f002:**
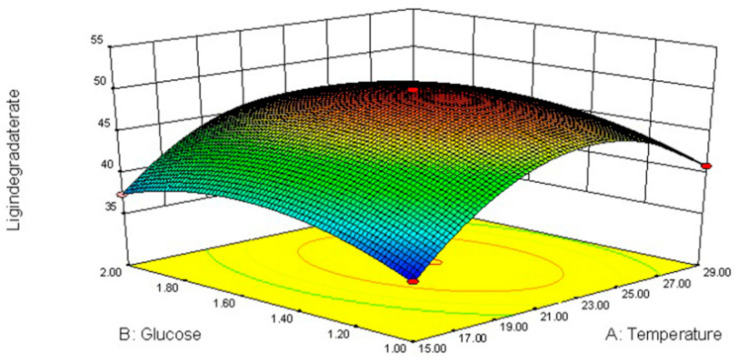
Response surface plot for optimum lignin degradation (21–23 °C and 1.5% glucose).

**Figure 3 jof-09-01156-f003:**
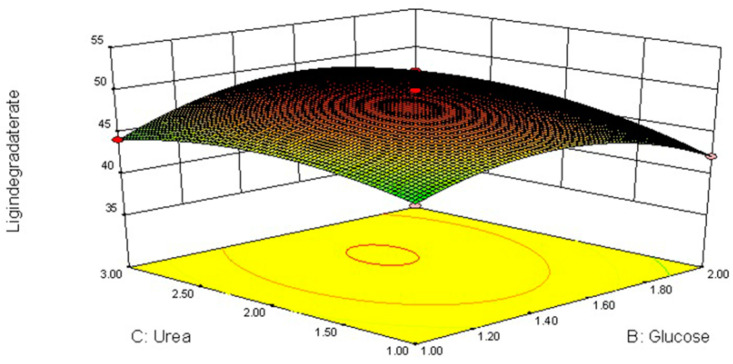
Response surface plot for optimum lignin degradation (2–2.5% urea and 1.4–1.6% glucose).

**Figure 4 jof-09-01156-f004:**
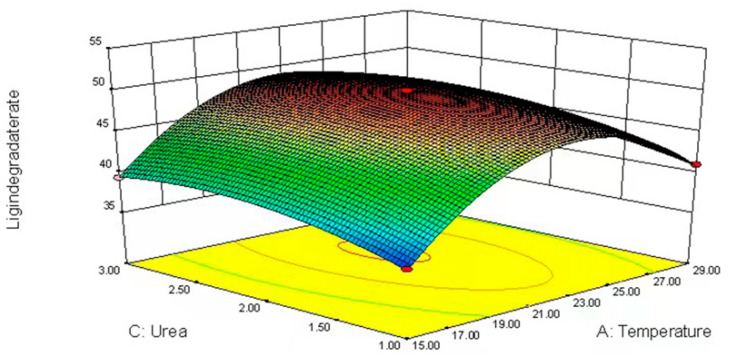
Response surface plot for optimum lignin degradation (21–23 °C and 2% urea).

**Table 1 jof-09-01156-t001:** Response surface methodology for rate of lignin degradation by *L. sajor-caju* treatment of highland barley straw.

Process Variables	Coded Level Variables		
	−1	0	1
Temperature (°C)	15 °C	22 °C	29 °C
Carbon (%)	1%	1.5%	2%
Nitrogen (%)	1%	2%	3%

**Table 2 jof-09-01156-t002:** The chemical compositions (dry matter (DM) basis) of highland barley straw incubated with various white rot fungi.

Item	Controls	*Lentinus sajor-caju*	*Pleurotus ostreatus*	*Phyllotopsis rhodophylla*	*Pleurotus djamor*	*Pleurotus eryngii*	*Pleurotus citrinopileatus*
CP	4.51 ± 0.13 ^E^	6.52 ± 0.25 ^A^	5.46 ± 0.10 ^C^	4.77 ± 0.14 ^D^	4.82 ± 0.07 ^D^	5.41 ± 0.05 ^C^	5.69 ± 0.17 ^B^
EE	1.36 ± 0.04 ^E^	2.27 ± 0.04 ^A^	1.83 ± 0.05 ^B^	1.67 ± 0.03 ^C^	1.55 ± 0.04 ^D^	1.80 ± 0.06 ^B^	1.78 ± 0.10 ^B^
Ash	6.79 ± 0.14 ^A^	4.59 ± 0.31 ^D^	5.49 ± 0.23 ^B^	5.49 ± 0.17 ^B^	5.31 ± 0.08 ^BC^	5.43 ± 0.04 ^B^	5.11 ± 0.03 ^C^
NDF	64.97 ± 1.62 ^A^	47.94 ± 0.29 ^F^	50.71 ± 0.97 ^E^	55.98 ± 0.59 ^B^	53.76 ± 0.40 ^C^	56.28 ± 0.31 ^B^	52.72 ± 0.59 ^D^
ADF	39.83 ± 0.66 ^A^	31.55 ± 0.67 ^E^	35.29 ± 1.20 ^C^	35.21 ± 2.10 ^C^	37.47 ± 0.18 ^B^	34.48 ± 0.13 ^CD^	33.39 ± 0.49 ^D^
ADL	19.01 ± 0.35 ^A^	10.21 ± 0.69 ^F^	12.92 ± 0.41 ^E^	15.96 ± 0.54 ^B^	13.61 ± 0.27 ^D^	14.51 ± 0.27 ^C^	15.53 ± 0.26 ^B^
Starch	0.99 ± 0.07 ^F^	1.81 ± 0.10 ^A^	1.35 ± 0.07 ^B^	1.12 ± 0.07 ^E^	1.26 ± 0.03 ^C^	1.16 ± 0.02 ^DE^	1.23 ± 0.01 ^CD^
SP	1.57 ± 0.04 ^G^	2.95 ± 0.06 ^A^	2.34 ± 0.06 ^B^	1.95 ± 0.02 ^E^	1.78 ± 0.02 ^F^	2.21 ± 0.03 ^D^	2.23 ± 0.03 ^C^
NPN	0.67 ± 0.02 ^F^	1.48 ± 0.03 ^A^	1.17 ± 0.02 ^B^	0.78 ± 0.01 ^E^	0.89 ± 0.01 ^D^	1.04 ± 0.01 ^C^	1.17 ± 0.02 ^B^
NDFIP	1.45 ± 0.05 ^A^	0.93 ± 0.01 ^E^	1.05 ± 0.02 ^D^	1.22 ± 0.02 ^C^	1.41 ± 0.02 ^A^	1.36 ± 0.06 ^B^	1.42 ± 0.04 ^A^
ADFIP	1.26 ± 0.01 ^A^	0.47 ± 0.01 ^F^	0.72 ± 0.02 ^E^	1.11 ± 0.02 ^C^	1.19 ± 0.02 ^B^	1.17 ± 0.06 ^B^	1.16 ± 0.02 ^B^

^A–G^ Within a row, means without a common superscript are significantly different (*p* < 0.001).

**Table 3 jof-09-01156-t003:** The Cornell Net Carbohydrate and Protein System (CNCPS) protein fractions (DM basis) of highland barley straw incubated with various white rot fungi.

Item	PA (%)	PB_1_ (%)	PB_2_ (%)	PB_3_ (%)	PC (%)
Controls	14.91 ± 0.47 ^G^	19.79 ± 0.44 ^D^	33.17 ± 0.50 ^C^	4.18 ± 0.97 ^D^	27.94 ± 0.88 ^A^
*Lentinus sajor-caju*	22.65 ± 0.74 ^A^	22.65 ± 0.89 ^B^	40.40 ± 2.10 ^A^	7.02 ± 0.30 ^A^	7.29 ± 0.45 ^F^
*Pleurotus ostreatus*	21.45 ± 0.58 ^B^	21.44 ± 0.88 ^C^	37.84 ± 1.05 ^B^	6.11 ± 0.64 ^B^	13.17 ± 0.35 ^E^
*Phyllotopsis rhodophylla*	16.38 ± 0.64 ^F^	24.46 ± 0.69 ^A^	33.51 ± 1.70 ^C^	5.32 ± 0.48 ^C^	20.33 ± 0.46 ^D^
*Pleurotus djamor*	18.36 ± 0.34 ^E^	18.64 ± 0.60 ^E^	33.69 ± 0.82 ^C^	4.56 ± 0.63 ^CD^	24.74 ± 0.67 ^B^
*Pleurotus eryngii*	19.12 ± 0.37 ^D^	21.62 ± 0.69 ^C^	34.08 ± 0.50 ^C^	4.65 ± 0.40 ^CD^	23.00 ± 0.73 ^C^
*Pleurotus citrinopileatus*	20.55 ± 0.52 ^C^	19.77 ± 0.74 ^D^	34.82 ± 1.81 ^C^	4.43 ± 0.70 ^CD^	24.27 ± 0.24 ^B^

^A–G^ Within a row, means without a common superscript are significantly different (*p* < 0.001).

**Table 4 jof-09-01156-t004:** CNCPS carbohydrate fractions of highland barley straw incubated with various white rot fungi.

Item	CHO/%DM	CA/%CHO	CB_1_/%CHO	CB_2_/%CHO	CC/%CHO	CNSC/%CHO
Controls	87.33 ± 0.24 ^B^	26.71 ± 0.69 ^G^	1.13 ± 0.09 ^F^	19.91 ± 1.50 ^B^	52.26 ± 1.03 ^A^	27.84 ± 0.71 ^E^
*Lentinus sajor-caju*	86.62 ± 0.54 ^C^	45.63 ± 0.37 ^A^	2.09 ± 0.13 ^A^	24.02 ± 1.95 ^A^	28.28 ± 1.77 ^F^	47.72 ± 0.29 ^A^
*Pleurotus ostreatus*	87.23 ± 0.27 ^B^	42.62 ± 1.18 ^B^	1.55 ± 0.09 ^C^	20.28 ± 2.16 ^B^	35.54 ± 1.10 ^E^	44.18 ± 1.17 ^B^
*Phyllotopsis rhodophylla*	88.07 ± 0.20 ^A^	37.07 ± 0.62 ^F^	1.27 ± 0.08 ^E^	18.18 ± 1.16 ^B^	43.48 ± 1.42 ^B^	38.34 ± 0.66 ^D^
*Pleurotus diamor*	88.07 ± 0.57 ^A^	39.23 ± 0.39 ^D^	1.37 ± 0.04 ^D^	18.58 ± 0.27 ^B^	38.20 ± 0.39 ^D^	40.81 ± 0.99 ^C^
*Pleurotus eryngii*	87.27 ± 0.26 ^B^	41.35 ± 0.48 ^C^	1.45 ± 0.02 ^D^	18.76 ± 0.46 ^B^	40.38 ± 0.21 ^C^	38.68 ± 0.55 ^D^
*Pleurotus citrinopileatus*	87.38 ± 0.25 ^B^	38.39 ± 0.21 ^E^	1.71 ± 0.03 ^B^	19.44 ± 0.15 ^B^	44.72 ± 0.91 ^B^	41.67 ± 0.81 ^C^

^A–G^ Within a column, means without a common superscript are significantly different (*p* < 0.001).

**Table 5 jof-09-01156-t005:** Box–Behnken design for variables and measured responses.

Run	Variables			Degradation (%)
	Temperature (°C)	Glucose (%)	Urea (%)	
1	0	1	1	46.90
2	−1	1	0	37.33
3	1	0	−1	41.00
4	−1	0	1	39.35
5	0	−1	−1	43.75
6	1	1	0	39.02
7	0	0	0	49.99
8	−1	0	−1	36.39
9	1	0	1	42.24
10	0	0	0	50.01
11	0	−1	1	44.17
12	−1	−1	0	35.23
13	0	0	0	49.86
14	1	−1	0	40.86
15	0	0	0	49.88
16	0	0	0	49.85
17	0	1	−1	42.16

**Table 6 jof-09-01156-t006:** In vitro fermentation products of control intact highland barely straw and highland barley straw pretreated with *L. sajor-caju*.

Item	Controls	*Lentinus sajor-caju*
IVDMD (%)	57.53 ± 0.19 ^B^	65.28 ± 0.64 ^A^
pH	7.05 ± 0.02	7.05 ± 0.02
Total VFA (mM)	83.21 ± 0.21 ^B^	89.52 ± 0.14 ^A^
Acetic acid (mM)	68.22 ± 0.10 ^B^	70.33 ± 0.12 ^A^
Propionic acid (mM)	17.38 ± 0.20 ^A^	16.79 ± 0.10 ^B^
Butyric acid (mM)	10.26 ± 0.09 ^B^	11.01 ± 0.03 ^A^
Ammonia nitrogen (mg/L)	65.13 ± 1.35 ^B^	86.67 ± 0.66 ^A^
Soluble protein (mg/L)	85.43 ± 0.60 ^B^	103.34 ± 0.89 ^A^
Microbial protein nitrogen (mg/L)	105.02 ± 1.14 ^B^	144.85 ± 1.67 ^A^

^A,B^ Within a column, means without a common superscript are significantly different (*p* < 0.001).

## Data Availability

Data are contained within the article.

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
