# Peer review of "Solid-State Fermentation with White Rot Fungi (Pleurotus Species) Improves the Chemical Composition of Highland Barley Straw as a Ruminant Feed and Enhances In Vitro Rumen Digestibility"

_jof, 2023, doi:10.3390/jof9121156_

Round 1

Reviewer 1 Report

Comments and Suggestions for Authors

I made several remarks/corrections.

The many abbreviations, especially already in the Abstract, make the manuscript very difficult to understand.

Use in the beginning of each chapter (Abstract, Introduction, Materal and Methods, Results, Discussion) at first the full names of the molecules with the abbreviation in parentheses.

The data in the Tables are hardly to understand.

Give a simple Table showing the initial values of lignin, cellulose and hemicelluloses and their contents after fungal attack.

Use current fungal names. You find them in Index Fungorum. Add the author names to each species.

Comments on the Quality of English Language

English can be improved.

Reviewer 2 Report

Comments and Suggestions for Authors

General comment:

1.      Latin words should be in italics, e.g. "in vitro "(line 50). Please check the whole text.

2.      Line 62 "incubated at 25 °C in rotary shakers "

Please use the preposition "on instead" "in ".

3.      Please write "Control "with a lowercase letter except when the sentence starts with this word.

4.      The authors should make sure that there are not any grammar and spelling mistakes.

5.      The authors should use standard units, e.g. g/l or mg/l instead of mg/dl.

Specific comments:

6.      Abstract, Lines 17-19:

"All white rot fungi significantly increased CP, 17 EE, Starch, SP and NPN content but decreased ash, NDF, ADF, ADL and ADFIP content. In addition, 18 P. sajor-caju treatment increased PA, PB2, PB3, CA, CB1, CB2, CNCPS …

The first time an abbreviation is used, it's important to spell out the full term and put the abbreviation in parentheses. Please write the "starch "in lowercase letters.

7.      Lines 58-59: "Agar plates were prepared using PDA and inoculated with 58 a 0.5 cm2 piece of the fungus at 25 °C for 7 days."

The sentence is not clear; please rewrite it. "…… of the fungus and incubated at 25 °C for 7 days."

8.      Authors should check and correctly write the names of the fungal strains used in the research. They should also list all the strains used in their research in subchapter 2.1, Fungal strains and spawn preparation.

9.      Results and Discussion

 In this chapter, the authors should compare their results with literature data, comment on the difference and explain what their study findings reveal that's unique or different from the existing literature. Authors should improve this chapter and avoid general comments , e.g. "Lignin decomposition in white rot fungi is highly regulated by nutrients and conditions (nutrient content and temperature) [29]. Furthermore, the degradation of lignin increases by optimizing the enitrogen source during fermentation. Protein and carbon source in the medium have key roles during fungal growth [30]. Concentrations of carbon sources in the fungal medium can have either positive or negative effects on the fungus. During fungal fermentation, the carbon source increases lignin degradation, which may stimulate increased synthesis of enzymes. However, effects of nitrogen source on lignocellulosic substrate depend not only on fungal physiology but also on other culture conditions [31]. Similarly, the other two factors also significantly affected mycelium production in the present study, emphasizing the need to optimize fungal fermentation conditions"

10.   Authors should describe the procedure for the determination of microbial protein nitrogen and volatile fatty acids in Materials and Methods.

Reviewer 3 Report

Comments and Suggestions for Authors

The manuscript content is interesting and several articles were published on the use soild-state fermentation with white root fungi to enhance agricultural wastes rumen digestibility in recent years.

The main problem of this manuscript is that it is not accurate. I just cheeked only few citations and they are not appropriate. Morover I checked just one  fungal name and I found that it are not the current name.

Several abbreviations are not explained and so the manuscript is very difficult to follow.

Several minor other points need attention:

Line 15 Pease check the current names of the fungi Pleurotus rhodophyllus Bres is now Phyllotopsis rhodophylla (Bres.) Singer

Line 14 spp. not in italic

Line 15 Pease check the current names of the fungi for example  Pleurotus rhodophyllus Bres is now Phyllotopsis rhodophylla (Bres.) Singer. Morover in the abstract do not add  the Authorities but only in the main text when the species is cited for the first time

Lines 17 and 18 Before to use the abbreviation you have to define them:  CP, 17EE, Starch, SP and NPN content but decreased ash, NDF, ADF, ADL and ADFIP content. In addition, 18P. sajor-caju treatment increased PA, PB2, PB3, CA, CB1, CB2, CNCPS

Line 21:In intro ????? Please do not use italic

Line 20-21 this sentence is not clear

Line 37 and 38 White rot fungi possess both cellulolytic and lignin degrading enzymes so they do not relase cellulose.

The article “Shabtay, A.; Hadar, Y.; Eitam, H.; Brosh, A.; Orlov, A.; Tadmor, Y.; Izhaki, I.; Kerem,Z. The potential of Pleurotus treated olive mill solid waste as cattle feed. Bio. Tech. 2009, 100, 6457-6464.” Does not demonstrate or affirms that white rot fungi degrade lignin more other microorganisms

The article “Sniffen, C.J.; O’Connor, J.D.; Soest, P.J.V.; Fox, D.G.; Russell, J.B. A Net Carbohydrate and protein system for evaluating cattle diets: II. Carbohydrate and protein availability. J. Anim. Sci.1992, 70,3562-3577.” Do not compare the capability of different white rot fungi do degrade lignin

In M & lines 53-55  Pleurotus rhodophyllus is not included. Here the Authorities are necessary in all the species names

Line 83-84 are repeated two lines later.

Line 92 the cited article dosen’t mention the boric acid-phoshate method

Lines 96- 121 It is not clear please explain what are proteins (PA, PB1, PB2, PB3, and PC) and carbohydrates (CA, CB1, CB2, and CC), you can refer to the cited article for the calculation method

Line 162 I don’t like Duncan test see DOI 10.7717/peerj.10387

Line 175 please specify what are the best results

Table 2 and others  SEM should be reported for each mean

Line 193 This sentence is not related to the previous one on beech logs. You have to delete in corn straw

Table 5 Please correct LIgnindegradaterate

Line 259 enitrogen?

Line 251 the pH was not affected

Line 308 digestibility not in italic

Comments on the Quality of English Language

The English needs to be improved

Round 2

Reviewer 1 Report

Comments and Suggestions for Authors

I again made some commentaries/corrections.

Generally, use current fungal names and authors. You find them in Index Fungorum.

Comments on the Quality of English Language

Reviewer 2 Report

Comments and Suggestions for Authors

The authors have been revised the manuscript according to the comments.

Author Response

The responses to reviewer #2:

The authors have been revised the manuscript according to the comments.

Response: thank you very much for the kind comments and suggestions. We acknowledge your comments and suggestions very much, which are valuable in improving the quality of our manuscript. Thank you for your time and patience.

Reviewer 3 Report

Comments and Suggestions for Authors

The manuscript is considerably improved after revision but numerous few points need to be addressed before publication.

Lines 14-16 the fungal names are not correctly written (please see the comments in the manuscript)

Lines 21-22 What are these abbreviations?

Lines 58-61. The Authorities should be not in italic and Singer. Has not the point. Please check all of them

Line 65 Do you take the agar plugs from the edge of the cultures or from the middle?

Tables Please delete the SEM and p columns or rows (you have already SEM after the means). Please also indicate more correctly under the tables:   “…….superscripts are significantly different” instead of “differ”. What is the level of significativity (0.01 or 0.001?)

I suggest to abbreviate highland barley straw (HBS)  (after the first citation)

Comments on the Quality of English Language
